# Effect of Monochloroacetic Acid on Properties of Carboxymethyl Bacterial Cellulose Powder and Film from Nata de Coco

**DOI:** 10.3390/polym13040488

**Published:** 2021-02-04

**Authors:** Pornchai Rachtanapun, Warinporn Klunklin, Pensak Jantrawut, Noppol Leksawasdi, Kittisak Jantanasakulwong, Yuthana Phimolsiripol, Phisit Seesuriyachan, Thanongsak Chaiyaso, Warintorn Ruksiriwanich, Suphat Phongthai, Sarana Rose Sommano, Winita Punyodom, Alissara Reungsang, Thi Minh Phuong Ngo

**Affiliations:** 1School of Agro-Industry, Faculty of Agro-Industry, Chiang Mai University, Chiang Mai 50100, Thailand; warinporn.k@cmu.ac.th (W.K.); noppol@hotmail.com (N.L.); jantanasakulwong.k@gmail.com (K.J.); yuthana.p@cmu.ac.th (Y.P.); phisit.s@cmu.ac.th (P.S.); thachaiyaso@hotmail.com (T.C.); suphat.phongthai@cmu.ac.th (S.P.); 2The cluster of Agro Bio-Circular-Green Industry (Agro BCG), Chiang Mai University, Chiang Mai 50100, Thailand; pensak.amuamu@gmail.com (P.J.); warintorn.ruksiri@cmu.ac.th (W.R.); 3Center of Excellence in Materials Science and Technology, Chiang Mai University, Chiang Mai 50200, Thailand; sarana.s@cmu.ac.th (S.R.S.); winitacmu@gmail.com (W.P.); 4Department of Pharmaceutical Sciences, Faculty of Pharmacy, Chiang Mai University, Chiang Mai 50200, Thailand; 5Plant Bioactive Compound Laboratory (BAC), Department of Plant and Soil Sciences, Faculty of Agriculture, Chiang Mai University, Chiang Mai 50200, Thailand; 6Department of Chemistry, Faculty of Science, Chiang Mai University, Chiang Mai 50200, Thailand; 7Department of Biotechnology, Faculty of Technology, Khon Kaen University, Khon Kaen 40002, Thailand; alissara@kku.ac.th; 8Research Group for Development of Microbial Hydrogen Production Process, Khon Kaen University, Khon Kaen 40002, Thailand; 9Academy of Science, Royal Society of Thailand, Bangkok 10300, Thailand; 10Department of Chemical Technology and Environment, The University of Danang–University of Technology and Education, Danang 550000, Vietnam; ntmphuong@ute.udn.vn

**Keywords:** bacterial cellulose, carboxymethyl bacterial cellulose, CMC*_n_*, carboxymethyl cellulose, CMC, monochloroacetic acid, MCA

## Abstract

Nata de coco has been used as a raw material for food preparation. In this study, the production of carboxymethyl cellulose (CMC) film from nata de coco and the effect of monochloroacetic acid on carboxymethyl bacterial cellulose (CMC*_n_*) and its film were investigated. Bacterial cellulose from nata de coco was modified into CMC form via carboxymethylation using various concentrations of monochloroacetic acid (MCA) at 6, 12, 18, and 24 g per 15 g of cellulose. The results showed that different concentrations of MCA affected the degree of substitution (DS), chemical structure, viscosity, color, crystallinity, and morphology of CMC*_n_*. The optimum treatment for carboxymethylation was found using 24 g of MCA per 15 g of cellulose, which provided the highest DS at 0.83. The morphology of CMC*_n_* was related to DS value; a higher DS value showed denser and smoother surface than nata de coco cellulose. The various MCA concentrations increased the mechanical properties (tensile strength and percentage of elongation at break) and water vapor permeability of CMC*_n_*, which were related to the DS value.

## 1. Introduction

Nata de coco is generally consumed as an ingredient in a dessert, which is produced by the action of *Acetobacter xylinum* (formerly known as vinegar bacterium) via acetic acid fermentation with coconut juice [1,2]. This bacterium is the most effective producer of cellulose, which uses carbon and nitrogen sources in a liquid medium [3]. Bacterial cellulose from *Acetobacter, Rhizobium, Agrobacterium, Aerobacter,* and *Pseudomonas* is integrated in the cell walls. Cellulose produced by *Acetobacter* has high purity without lignin and hemicellulose [3]. Many research projects have studied the production of bacterial cellulose that is converted to carboxymethyl cellulose, such as carboxymethylated-bacterial cellulose for copper and lead ion removal [4] and artificial blood vessels for microsurgery [5]. However, the production of CMC film from nata de coco (CMC*_n_*) is limited.

Cellulose forms the main structure of plant cell walls, which are linear chains of glucose units combined with β-glycosidic bonds between the C-4 of one sugar unit and the anomeric C-1 of the second sugar unit. Cellulose is held together by lignin and hemicellulose. Cellulose does not dissolve readily in hot/cold water nor in common solvents [6]. To use cellulose in the food industry, it must be converted to useful derivatives [7]. Cellulose derivatives, such as sodium carboxymethyl cellulose (CMC), are used in a variety of industries. CMC is a copolymer of two units, β-*D*-glucose and β-*D*-glucopyranose 2-O-(carboxymethyl)-monosodium salt, which are combined together through a β-1, 4-glycosidic bond. The substitution of the hydroxyl groups with the carboxymethyl groups is slightly supereminent at C-2 of the glucose position [8]. The quality of CMC to be implemented depends on the type of application, such as technical, semi-purified, or purified CMC. Purified CMC is a white to creamy yellow color, tasteless and odorless, and is a free-flowing powder [9]. CMC is an important cellulose derivative used in many industrial applications such as the food industry, cosmetics, pharmaceuticals, detergents, textiles, ceramics, packaging films, etc. [6,10] CMC is prepared by etherification of the hydroxyl groups with sodium monochloroacetate (NaMCA) in aqueous NaOH. The first step in the carboxymethylation process is an equilibrium reaction between NaOH and the hydroxyl groups of the cellulose, as shown in Equations (1) and (2).
CLL–OH + NaOH → CLL−ONa + H_2_O(1)
CLL–ONa + Cl−CH_2_– CO−ONa → CLL−O−CH_2_−COONa + NaCl(2)

(Sodium monochloroacetate) (Carboxymethylcellulose)

The second step is the actual formation of the carboxymethyl group through substituting NaMCA to create CMC through a side reaction with NaOH to form sodium glycolate, as shown in Equation (3).
NaOH+ Cl−CH_2_−CO−ONa→ HO−CH_2_−CO−ONa + NaCl(3)

(Sodium hydroxide) (Sodium glycolate)

There are many factors that determine CMC characteristics, including CMC concentration [11], NaMCA concentration [9], NaOH concentration [6,12,13,14], temperature [6], and reaction time [15]. However, plant cellulose causes deforestation and environmental problems. Therefore, many research projects have reported that CMC can be synthesized from agricultural wastes such as sugar beet pulp [16], cavendish banana pseudo-stems [9], sago waste [17], papaya peels [18], *Mimosa pigra* peels [6], durian rinds [12], mulberry paper [19], asparagus stalk end [14], and bacterial cellulose [20].

Therefore, the aim of this study was to determine the effects of monochloroacetic acid (6, 12, 18, and 24 g/15 g cellulose) in CMC synthesis on the degree of substitution, viscosity, chemical structure, morphology, and solubility. The effects of monochloroacetic acid on the mechanical properties and water vapor permeability (WVP) of CMC*_n_* films were also investigated.

## 2. Materials and Methods

### 2.1. Materials

Nata de coco was obtained from the Watphleng District, Ratchaburi province, Thailand; magnesium sulfate from Prolabo, England; and commercial grade citric acid from Thai Roong Ruang Sugar Group, Thailand. Sodium hydroxide and glacial acetic acid were purchased from Northern Chemicals and Glasswares, Chiang Mai Province, Thailand. Hydrochloric acid and sodium chloride were obtained from Merck, Germany, and monochloroacetic acid (MCA) from Sigma-Aldrich, St. Louis, MO, USA, with m-cresol purple, indicator grade, from Himedia, India. Ethanol and methanol were obtained from the distillation of recycled packaging, Faculty of Agro-Industry, Chiang Mai University, Thailand. All chemical reagents were of analytical grade.

### 2.2. Extraction of Bacterial Cellulose

First, the nata de coco was sliced and boiled in a pot 5 times, dried in a hot air oven (60 °C, 24 h) and ground into powder with a mesh size below 35. It was then stored in a polyethylene (PE) bag.

### 2.3. Synthesis of Carboxymethyl Cellulose from Bacterial Cellulose (CMC_n_)

CMC*_n_* was synthesized following the process reported by Rachtanapun et al. [18] and Rachatnapun and Rattanapanone [6]. A total of 15 g of bacterial cellulose powder, 100 mL of NaOH solution (30/100 mL), and 450 mL of isopropanol were mixed in a beaker for 30 min. Various amounts of MCA (6, 12, 18, and 24 g/15 g of cellulose) were added to start the carboxymethylation. The mixture was continuously stirred at 55 °C for 30 min and then covered with aluminum foil. The mixture was placed in a 55 °C oven for 3.5 h. After heating, the solution was separated into two phases. The liquid phase was removed. The solid phase was suspended in methanol, neutralized with acetic acid (80 mL/100 mL), and then filtered using a Buchner funnel. The final product was washed three times with 70% (*v*/*v*) ethanol and then washed with methanol again. The obtained carboxymethyl bacterial cellulose (CMC*_n_*) was dried in a 55 °C oven for 12 h and kept in dry place.
(4)$$Yield\;of\;CMC\;\left(\%\right)=\frac{Weight\;of\;CMC\;\left(g\right)}{Weight\;of\;cellulose\;\left(g\right)}\times 100$$

### 2.4. Color Characteristics

The color characteristics of cellulose from nata de coco and CMC*_n_* were evaluated with a Color Quest XE Spectrocolorimeter (Hunter Lab, Shen Zhen Wave Optoelectronics Technology Co., Ltd., Shenzhen, China) to express the CIELAB color as three values: *L** for the lightness, ranging from blackness (0) to whiteness (100); *a**, ranging from greenness (−) to redness (+); and *b**, ranging from blueness (−) to yellowness (+). The total color differences (Δ*E*) consider the comparisons between the *L**, *a**, and *b** values of the sample and standard and are calculated by Equation (5) [14].
(5)$$\Delta E=\sqrt{{\left(L_{Standard}^{*}-L_{Sample}^{*}\right)}^{2}+{\left(a_{Standard}^{*}-a_{Sample}^{*}\right)}^{2}+{\left(b_{Standard}^{*}-b_{Sample}^{*}\right)}^{2}}$$

The whiteness index (*WI*) was also calculated with Equation (6) to represent the degree of whiteness in samples [12].
(6)$$WI=\sqrt{\left(100-L^{{*}^{2}}\right)+a^{{*}^{2}}+b^{{*}^{2}}}$$

### 2.5. CMC_n_ Film Preparation

We dissolved 3.0 g of CMC*_n_* in 300 mL of distilled water at 80 °C for 15 min to obtain film-forming solutions, and glycerol (30% *w*/*w*) was added. Then, the film-forming solutions were cast onto plates (20 cm × 15 cm). The plate was dried at room temperature for 48 h and CMC*_n_* film was obtained. The films were peeled off the plates, kept in polyethylene bags, and placed in a desiccator. The film was cut for property testing. Mechanical properties testing (tensile strength and elongation at break) required specimens of 1.5 cm × 14 cm rectangular strips (ASTM, D828-80a, 1995a). Circular specimens with an 8 cm diameter were used for water vapor transmission rate (WVTR) testing. The thickness of the films was measured using a micrometer (model GT- 313-A, Gotech testing machine Inc., Taichung Industry Park, Taichung City, Taiwan). Measurements were taken at five different locations on each sample and the average values were used in calculating tensile strength and water vapor permeability (WVP). All mechanical tests were performed with 10 replications.

### 2.6. The Degree of Substitution (DS) of CMC_n_

The degree of substitution was calculated with the United State Pharmacopeia (The USP) XXIII method and carboxymethyl and sodium carboxymethyl groups at C2, C3, and C6. The method included two steps: titration and residue on ignition as described by Rachtanapun and Rattanapanone [6]. The reported DS values were the average of three determinations. The *DS* was calculated using Equation (7):*DS* = *A* + *S*(7)
where *A* is the degree of substitution of carboxymethyl acid and *S* is the degree of substitution of sodium carboxymethyl.

During the titration step, 1 g of CMC*_n_* was weighed and placed in an Erlenmeyer flask (500 mL). After the CMC*_n_* stood for 5 min with intermittent shaking, 5 drops of m-cresol and 15 mL of hydrochloric solution (0.1 M) were added and shaken. If the solution was still violet, hydrochloric solution was added until the solution turned yellow. Then the solution was back-titrated with sodium hydroxide (0.1 M). At the endpoint, the solution turned violet.

The next step was residue on ignition. A crucible was placed in a 100 °C oven for 1 h and kept in a desiccator for 30 min. A 0.1000 g sample of CMC*_n_* was weighed (weighing apparatus, AR3130, Ohaus Corp. Pine, Brook, NJ, USA) and added to a crucible. The crucible containing CMC*_n_* was ignited using a hot plate until black residue was obtained. Then, sulfuric acid was used to damp it. It was heated until white fumes volatilized and was ignited at 600 °C. White residue was obtained and placed in desiccators. The percentage of residue on ignition was calculated using Equation (8):(8)$$The\;percentage\;of\;ash\;remainingafter\;ignition=\;\frac{Weight\;of\;residue\;content}{Weight\;of\;CMC\;content}\times 100$$
where M is the mEq of base required from the titration to the end point and C is the percentage of ash remaining after ignition.

### 2.7. Fourier Transform Infrared Spectroscopy (FTIR)

The functional groups of the bacterial cellulose powder from nata de coco and CMC*_n_* powder samples were determined by using an infrared spectrophotometer (Bruker, Tensor27, Berlin, Germany). The transmission was measured at a wave range of 4000–400 cm^−1^ [14].

### 2.8. Viscosity

The viscosities of the bacterial cellulose and CMC*_n_* were measured using a Ripid Visco Analyzer (Model: RVA-4, Warriewood, Australia). We dissolved 3.0 g of bacterial cellulose and the CMC*_n_* sample in 25 mL of distilled water by stirring at 80 °C for 10 min. The speed was first set at 960 rpm for 10 s, while the varied temperature was set to 30, 40, and 50 °C for 5 min, and then the speed was set to 160 rpm [6].

### 2.9. Scanning Electron Microscopy (SEM)

The morphology of the bacterial cellulose in the CMC*_n_* powder was analyzed through scanning electron microscopy (SEM) (Phillip XL 30 ESEM, FEI Company, Hillsboro, OR, USA). The CMC*_n_* powder sample was coated with gold through evaporation in a high vacuum or spray-coating in a low vacuum. A gold coating (sputter coater) on the sample prevented the accumulation of a static electric field due to electron irradiation. The acceleration voltage was 15 kV with 2000x.

### 2.10. Solubility

The percentage of solubility of the CMC*_n_* films was measured using a method modified from Rachtanapun and Rattanapanone [6]. First, the CMC*_n_* films were dried at 105 °C for 24 h, kept in desiccators for 24 h, weighed at 0.2000 g initial dry weight (*W_i_*), and suspended in 50 mL of distilled water with shaking at 500 rpm for 15 min. Then, they were poured onto filter papers (Whatman, No. 93) and dried at 105 °C for 24 h. The final dry weight (*W_f_*) was obtained. The percentage of soluble matter (%*SM*) of films was calculated using Equation (9):(9)$$\%SM=\frac{\left(W_{i}-W_{f}\right)}{W_{i}}\times 100$$

### 2.11. Mechanical Properties

Mechanical properties (tensile strength (TS) and percentage elongation at break (EB)) were measured using a Universal Testing Machine Model 1000 (HIKS, Selfords, Redhill, England) according to the ASTM Method (ASTM, D882-80a, 1995a). CMC*_n_* films were preconditioned (27 ± 2 °C, 65 ± 2% relative humidity (%RH), 24 h) before testing. The precondition was described in detail elsewhere [18]. The initial grip separation and crosshead speed were set at 100 and 20 mm/min, respectively. The TS value was calculated by dividing the maximum load with the initial cross-sectional area of the specimen. The EB value was calculated as the percentage of change of the initial gauge length of a specimen (100 mm) at the point of a sample failure (ASTM, D828-80a, 1995a). All mechanical tests were performed with 10 replications.

### 2.12. Water Vapor Transmission Rate (WVTR)

The water vapor transmission rate (WVTR) of the CMC*_n_* films was measured using the ASTM method (ASTM, E96-93, 1993). Cups containing ten grams of dried silica gel were covered with the specimens and sealed with paraffin wax. The sealed cups were weighed and kept at 25 °C in a desiccator with a saturated solution of sodium chloride (NaCl) to provide 75% RH. Then, the cups were re-weighed daily for 10 days. The water vapor transmission rate (*WVTR*) of the films was measured based on the weight gain of the cups and calculated using Equation (10):(10)$$WVTR=\frac{slope}{film\;area}$$
where slope is the slope of linear equation of time (*y*-axis) versus weight gain (*x*-axis).

The water vapor permeability (*WVP*) (g m^−2^ mmHg^−1^ day^−1^) was calculated as Equation (11):(11)$$WVP=\frac{WVTRxL}{\Delta P}$$
where *L* is the mean film thickness (mm) and Δ*P* is the partial water vapor pressure difference (mmHg) across the two sides of the film specimen.

### 2.13. Statistical Analysis

Statistical data were analyzed by one-way analysis of variance (ANOVA) using SPSS software version 16.0. Duncan’s multiple range test was employed to evaluate significant differences among the treatments (*p* < 0.05). All measurements were analyzed in triplicate. The results are presented as the average value ± standard deviation. The error bars for some data points may overlap with the mean values.

## 3. Results and Discussion

### 3.1. Degree of Substitution (DS) and Percent Yield of CMC_n_

The CMC obtained by etherification using sodium monochloroacetic acid and sodium hydroxide was in the range of 0.4–1.3 (Waring and Parsons, 2011). When DS value was less than 0.4, the CMC was able to swell, but was still insoluble. By contrast, when the DS value was over 0.4, the CMC was soluble [9,15]. The CMC was found with DS values of 0.20 and 0.83. The effects of MCA levels on the DS of CMC are shown in Figure 1. The DS of CMC increased with the elevation of MCA levels. A maximum DS of 0.83 was obtained with 24 g/15 g cellulose of MCA. At high levels of MCA, the MCA molecules were more available and substitution in molecules of cellulose was easier [16,20]. A similar result was reported regarding the DS value of CMC from sugar beet pulp [16], banana pseudo-stems [9], *Mimosa pigra* peels [6], durian rinds [12], and asparagus stalk end [14].

The cellulose converted to CMC in this synthesis process yielded two main products. The first product was produced from the reaction of cellulose hydroxyl reaction with MCA to obtain CMC*_n_* and the second product was formed from a by-product (sodium glycolate) [7]. The percentage yield of bacterial cellulose increased when the MCA levels increased (Figure 2). The increase in the percent yield was correlated with an increased DS value (Figure 1). This result is consistent with the percentage yield of CMC from cavendish banana pseudo-stems, which increased with increasing DS [6,9,12].

### 3.2. Solubility

The effect of MCA levels on the solubility of CMC*_n_* powders is shown in Figure 3. The solubility of CMC*_n_* increased as MCA levels elevated from 6 to 24 g/15 g of cellulose. At low levels of MCA (6 and 12 g/15 g of cellulose), the ability of CMC to immobilize water in a system was also low [6]. At high levels of MCA (18 and 24 g/15 g of cellulose), more carboxymethyl groups of cellulose acted as hydrophilic groups. Therefore, CMC showed improved solubility with increasing DS.

### 3.3. Color

Color was measured to determine changes caused by the carboxymethylation reaction. The main effect on color values of CMC*_n_* was an increase in *L** and *b** values as MCA levels increased up to 24 g/15 g of cellulose (Table 1). For MCA levels below 6 g/15 g of cellulose, the result was a decrease in the *a** value. This indicated the use of MCA with levels ranging from 12 to 24 g/15 g of cellulose. The results for ΔΕ showed the same effect on *a** values. The WI had the same tendency as the *L** and *b** values. The color changes might confirm the carboxymethylation reactions [12].

### 3.4. Fourier Transform Infrared Spectroscopy (FTIR)

FTIR confirmed the broad absorption bands of representative spectra for bacterial cellulose and CMC*_n_* synthesized with 24 g/15 g of cellulose, as shown in Figure 4. Typical bands detected in the region of 1630–900 cm^−1^ were assigned to cellulose. The 1633 cm^−1^ peak was the valence vibration of the cellulose water molecules [6]. In this study, cellulose and CMC*_n_* had similar functional groups, such as a hydroxyl group (–OH stretching) at 3200–3600 cm^−1^, a methyl group (–CH_2_ bending) at 1420 cm^−1^, a carbonyl group (C=O stretching) at 1600 cm^−1^, and ether groups (–O– stretching) at 1021 cm^−1^ [6,14,18,21]. In the CMC*_n_*, the carbonyl group and the methyl group increased on the representative spectra, but the hydroxyl group decreased. The results indicated carboxymethyl substitution of the cellulose molecules [6].

### 3.5. Effect of Various MCA Levels on Viscosity of CMC_n_

The viscosity of CMC solutions mainly depends on the CMC concentration [11], NaMCA concentration [9], NaOH concentration [14], and temperature [6]. The effects of various MCA levels on the viscosity of CMC*_n_* synthesized with 6, 12, 18, and 24 g/15 g of cellulose are shown in Figure 5. At the same temperature, the viscosity of the CMC*_n_* solution increased with increasing MCA levels, as more carboxymethyl groups, acting as hydrophilic groups, substituted for hydroxyl groups of cellulose. The viscosity of the CMC*_n_* solution decreased when the temperature increased, as CMC*_n_* can cause reductions in the cohesive force while simultaneously increasing the rate of molecular interchange [6,22]. The increase in viscosity was also correlated with an increased DS value [6].

### 3.6. Scanning Electron Microscopy of CMCn Powder

The scanning electron micrographs of bacterial cellulose from nata de coco and CMC*_n_* powder with 6, 12, 18, and 24 g/15g of cellulose MCA levels are shown in Figure 6a–e. The morphology of bacterial cellulose from nata de coco (Figure 6a) included the appearance of a number of small fibers. However, the microstructure of CMC*_n_* dramatically changed when the MCA levels increased [12]. The morphology of CMC*_n_* with 6 and 12 g/15g of cellulose MCA levels showed a rough surface. When MCA levels were further increased to 18–24 g/15 g of cellulose, the surface was more compact, denser, and smoother. The morphology of the CMC*_n_* powder with MCA levels also correlated with the DS value due to a carboxymethylation reaction, resulting in a surface morphology change from fibrous to smooth. This result is similar to carboxylmethyl cellulose from sago waste [17].

### 3.7. Water Vapor Permeability (WVP)

CMC solutions with less than 12 g/15 g of cellulose of MCA could not form films due to the lower DS values and solubility. The effect of MCA levels on water vapor permeability (WVP) of CMC*_n_* films is shown in Figure 7. The films that were cast using 24 g/15 g of cellulose of MCA showed the highest WVP values because the cellulose was converted to CMC*_n_*, which caused a higher polarity related to the hydrophilic nature of the film [6]. The results are in agreement with CMC from *Mimosa pigra* peel films since the hydrophilic nature similarly increased with increasing carboxymethyl groups [18,23,24]. The WVP of CMC*_n_* is directly related to the DS value.

### 3.8. Tensile Strength (TS) and Elongation at Break (%EB)

The tensile strength values of CMC films with various MCA levels are shown in Figure 8a. The tensile strength increased as the MCA level increased (18–24 g/15 g of cellulose). This is because TS values are related to increasing DS values due to the substitution of carboxymethylation causing an increase in the intermolecular force [9]. These results are similar to tensile properties found in CMC from *Mimosa pigra* peel [6].

The percentage of elongation at break (EB) of CMC*_n_* films with various MCA levels of synthesized CMC*_n_* are shown in Figure 8b. The elongation at break of CMC*_n_* films increased with increasing MCA levels (18 and 24 g/15 g of cellulose) due to higher MCA levels decreasing the crystallinity and increasing the flexibility of the cellulose structure [6].

## 4. Conclusions

Bacterial cellulose was successfully extracted from nata de coco. Then, the bacterial cellulose was modified to CMC*_n_* through carboxymethylation with various MCA concentrations (6, 12, 18, and 24 g/15 g of cellulose). The effects of the CMC film preparations and MCA concentrations on mechanical properties, WVP, viscosity, and solubility of the films were studied. The results were highly related to DS values. The highest DS values and mechanical properties were obtained from CMC films synthesized with 24 g/15 g of cellulose of MCA. In addition, the elongation at break was also related to DS values.

## Figures and Tables

**Figure 1 polymers-13-00488-f001:**
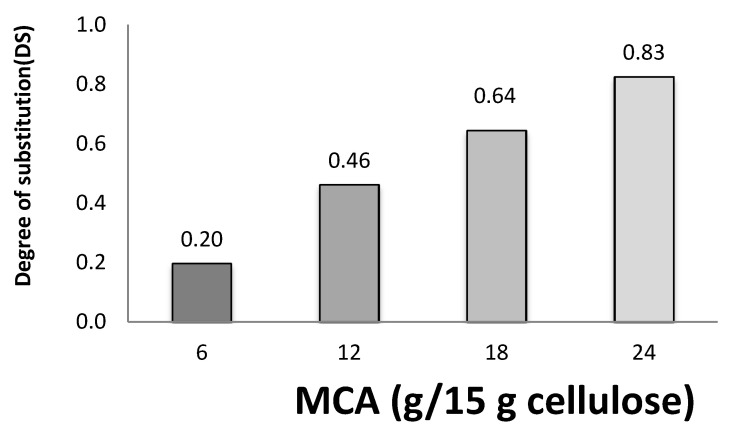
Effects of monochloroacetic acid (MCA) levels on degree of substation (DS) of Carboxymethyl Bacterial Cellulose from Nata de Coco (CMC*_n_*).

**Figure 2 polymers-13-00488-f002:**
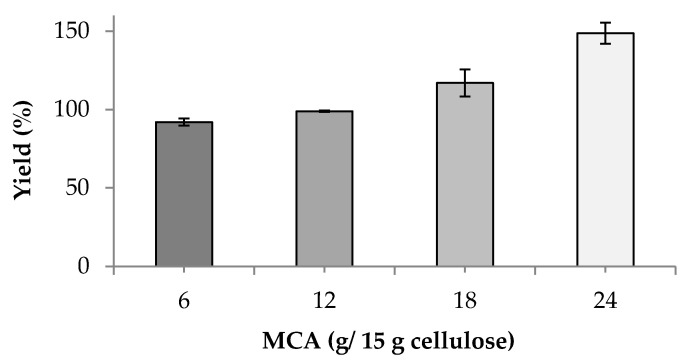
Effects of MCA levels on the percent yield of CMC*_n_*.

**Figure 3 polymers-13-00488-f003:**
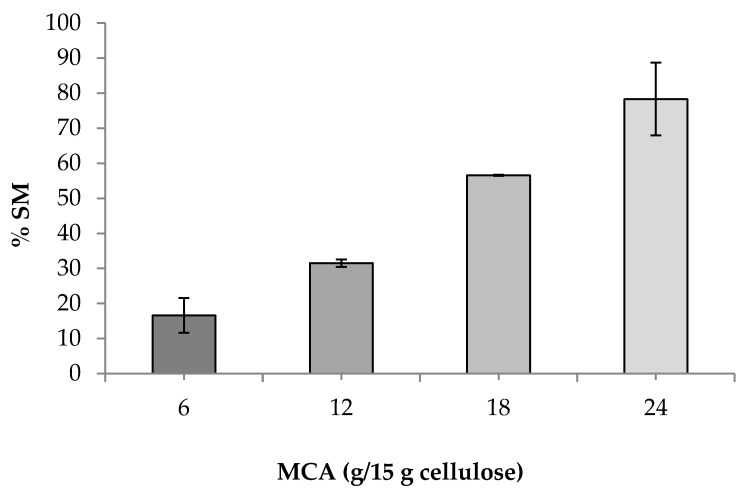
Effects of MCA levels on solubility of CMC*_n_*.

**Figure 4 polymers-13-00488-f004:**
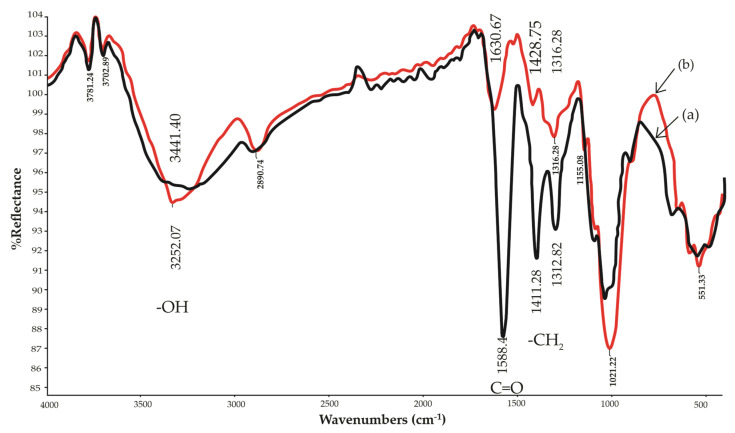
Fourier transform infrared spectroscopy of (**a**) bacterial cellulose from nata de coco and (**b**) CMC*_n_* synthesized with 24 g MCA/15 g cellulose.

**Figure 5 polymers-13-00488-f005:**
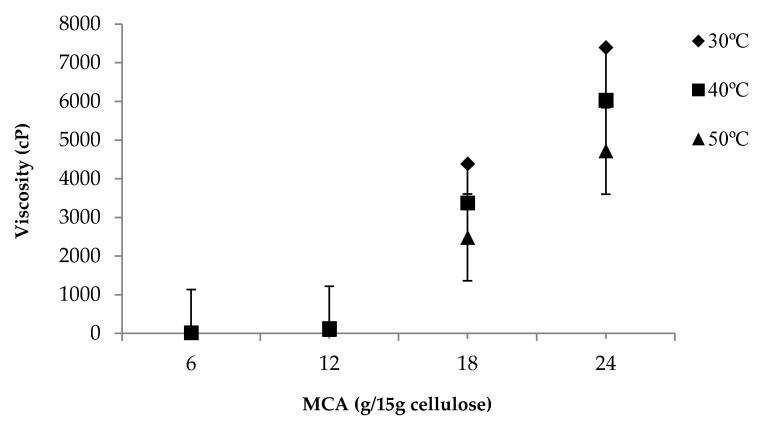
Differential scanning calorimetry of cellulose from asparagus stalk end and CMC*_n_* synthesized with various NaOH concentrations.

**Figure 6 polymers-13-00488-f006:**
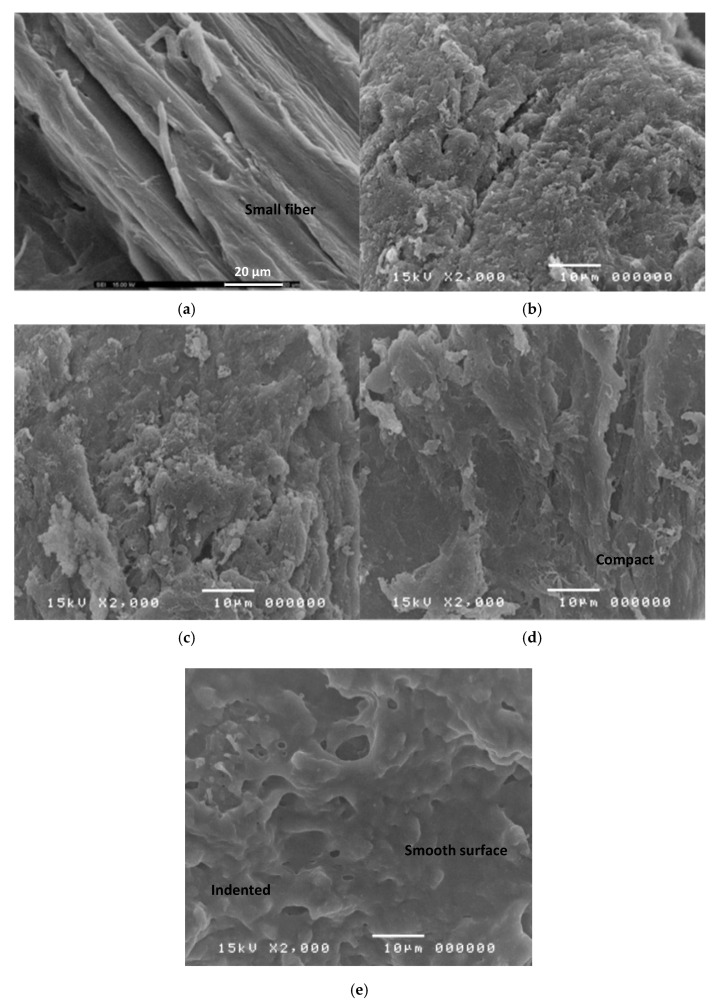
Scanning electron micrographs of (**a**) cellulose from nata de coco and CMC*_n_* powder with (**b**) MCA 6 g/15 g of cellulose, (**c**) MCA 12 g/15 g of cellulose, (**d**) MCA 18 g/15 g of cellulose, and (**e**) MCA24 g/15g of cellulose.

**Figure 7 polymers-13-00488-f007:**
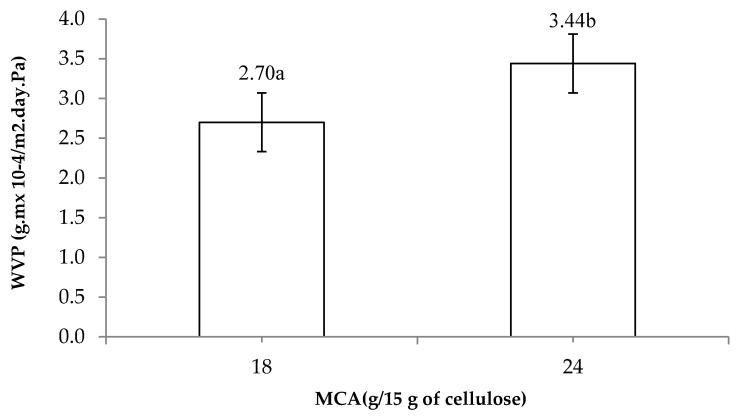
Effects of MCA levels on water vapor permeability (WVP) of CMC*_n_* film. The different letter, e.g., ‘a’ and ‘b’ are statistically different (*p* < 0.05).

**Figure 8 polymers-13-00488-f008:**
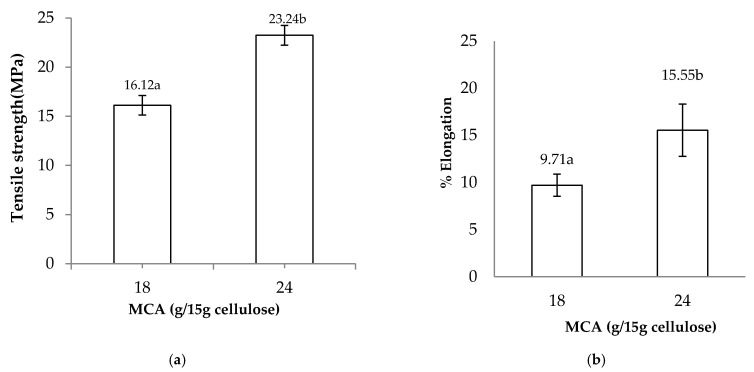
Effects of various MCA levels (18 and 24 g/15 g of cellulose) on (**a**) tensile strength and (**b**) percent elongation at break of CMC*_n_* films. The different letter, e.g., ‘a’ and ‘b’ are statistically different (*p* < 0.05).

**Table 1 polymers-13-00488-t001:** Color values of bacterial cellulose from nata de coco and CMC*_n_* synthesized with various amounts of MCA.

MCA (g/15 g of Cellulose)	*L**	*a**	*b**	Δ*Ε*	YI	WI
6	67.99 ^a^	5.34 ^a^	14.65 ^a^	29.17 ^a^	30.81 ^a^	64.33 ^a^
12	75.22 ^b^	3.34 ^a, b^	15.10 ^a^	22.92 ^a^	28.68 ^a^	70.79 ^b^
18	77.57 ^a^	2.95 ^b^	15.85 ^a^	22.28 ^a^	29.58 ^a^	71.54 ^b^
24	78.00 ^b^	2.37 ^c^	17.28 ^a^	22.15 ^b^	31.67 ^a^	71.91 ^b^

Note: Obtained in Duncan’s test (*p* < 0.05). *L** = lightness, *a** = redness, *b** = yellowness, Δ*E* = total color difference, YI = yellowness index, and WI = whiteness index. Different superscript letters (e.g., ‘a’ and ‘b’) indicate the statistical significance among treatments (*p* < 0.05).

## Data Availability

The data presented in this study are available on request from the corresponding author.

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
