# Peer review of "Effect of Monochloroacetic Acid on Properties of Carboxymethyl Bacterial Cellulose Powder and Film from Nata de Coco"

_polymers, 2021, doi:10.3390/polym13040488_

Round 1

Reviewer 1 Report

Submitted manuscript entitled “Effect of Monochloroacetic Acid on Properties of Carboxyme-2 thyl Bacterial Cellulose Powder and Film from Nata de Coco” described  production of carboxymethyl cellulose (CMC) film from Nata de coco.

The authors in this work use many complementary methods, but there is no basic thermal analysis (DSC and TGA). If the films had to have application in industry, the analysis would be necessary. It is about the storage conditions and the use of these materials at higher temperatures. The authors write that they added glycerol while producing the film. It may improve the mechanical properties, but doesn't it reduce the temperature stability range of the system? Please compare whether the obtained films have mechanical properties comparable to similar systems described in the literature. The article is written in a clear way.

Author Response

RESPONSES TO REFEREE #1’S COMMENTS:

The authors in this work use many complementary methods, but there is no basic thermal analysis (DSC and TGA). If the films had to have application in industry, the analysis would be necessary. It is about the storage conditions and the use of these materials at higher temperatures. The authors write that they added glycerol while producing the film. It may improve the mechanical properties, but doesn't it reduce the temperature stability range of the system? Please compare whether the obtained films have mechanical properties comparable to similar systems described in the literature. The article is written in a clear way.

Thank you very much for this insightful comment about thermal analysis. In general, the higher content of crystalline polymer in the system, the higher thermal stability and mechanical properties of films were obtained, which was evidenced by the results of tensile test. We totally understand your point of view. However, thermal analysis is out of our scope and takes some time to evaluate. Therefore, we are unable to analyze within 5 days due to the laboratory safety practices of our university during the COVID-19 pandemic.

Reviewer 2 Report

The article entitled "Effect of Monochloroacetic Acid on Properties of Carboxymethyl Bacterial Cellulose Powder and Film from Coconut Cream" is very well written and structured, presenting good results and characterizations, and should be accepted in this excellent journal after minor adjustments, discussed below.
1. The term “nata de coco” should be replaced by coconut cream.

2.  In item 2.1 Materials, you need to identify at least the purity or concentration of the reagents used.

3. Some characterizations have incomplete data.

4. In Figures (1, 2) replace MCA (g / 15 g cellulose) per MCA (g / 15 g cellulose)

5. The authors could propose a correlation between the substitution grade and degree of substitution. And even it can be done for solubility and other properties.
6. For the color parameters identified, a trend when the amount of MCA increases, but the parameters a * and YI does not occur, why?

7. Check the use of the term scissoring in the FTIR assignments.

8. It would be ideal to describe the main bands in the cellulose FTIR and then describe the changes that have occurred through the modification

9. Check and Correct the title in Figure 5.10. If authors can add results from thermal analysis and NMR it would greatly enrich the publication.

Author Response

RESPONSES TO REFEREE #2’S COMMENTS:

The article entitled "Effect of Monochloroacetic Acid on Properties of Carboxymethyl Bacterial Cellulose Powder and Film from Coconut Cream" is very well written and structured, presenting good results and characterizations, and should be accepted in this excellent journal after minor adjustments, discussed below.

Thank you very much for this insightful comment about our paper.

  1. The term “nata de coco” should be replaced by coconut cream.        

Nata de coco, also marketed as coconut gel, is a chewy, translucent, jelly-like food produced by the fermentation of microorganisms and coconut water which is quite different from coconut cream. Therefore, we prefer to use Nata de coco.

  1. In item 2.1 Materials, you need to identify at least the purity or concentration of the reagents used.

The purity of the reagents was added in lines 102-103.

  1. Some characterizations have incomplete data.

            All characteristics have been double checked.

  1. In Figures (1, 2) replace MCA (g / 15 g cellulose) per MCA (g / 15 g cellulose)

The texts were revised.

  1. The authors could propose a correlation between the substitution grade and degree of substitution. And even it can be done for solubility and other properties.

Thank you very much for a good comment. We believe we tried to tell the correlation between DS and other properties.  

Therefore, CMC showed improved solubility with increasing DS. In line 258-259. 

The increase in viscosity was also correlated with an increased DS value [6]. In line 298.

The WVP of CMCn is directly related to the DS value. In line 324-325.

This is because TS values are related to increasing DS values due to the substitution of carboxymethylation causing an increase in the intermolecular force. In line 331-332.

  1. For the color parameters identified, a trend when the amount of MCA increases, but the parameters a * and YI does not occur, why?

An a* and YI values of CMCn synthesized with 24 g of MCA per15 g of cellulose) were statistically different from the other. These parameters also changed according to the different concentration of MCA during the synthesis. Moreover, the superscript letters were added in a* and YI values.  

  1. Check the use of the term scissoring in the FTIR assignments.

The word “scissoring” was amended to “bending” in line 281.

  1. It would be ideal to describe the main bands in the cellulose FTIR and then describe the changes that have occurred through the modification.

The typical bands for cellulose were described further in lines 277-279

  1. Check and Correct the title in Figure 5.10. If authors can add results from thermal analysis and NMR it would greatly enrich the publication.

Thank you very much for a good comment. The title of figure 5 has been checked and revised. Thermal analysis is out of our scope and takes some time to evaluate. Therefore, we are unable to analyze within 5 days due to the laboratory safety practices of our university during the COVID-19 pandemic.

We believe we have attended to all the corrections suggested. We have also made small changes to our text to make some other points clearer. Thank you for taking the time and energy to help us improve the paper.
